# Perceiving threat in others: The role of body morphology

**Terence J. McElvaney** [ID]◉*, **Magda Osman, Isabelle Mareschal**◉

Department of Biological and Experimental Psychology, School of Biological and Chemical Sciences, Queen Mary University of London, London, United Kingdom

◉ These authors contributed equally to this work.
* t.mcelvaney@qmul.ac.uk

**Data Availability Statement:** The data for Experiments 1 & 2 are both publicly available on the Open Science Framework. Experiment 1: https://osf.io/mwdje/ Experiment 2: https://osf.io/v3fz6/ These links are provided in the submitted manuscript.

## Abstract

People make judgments of others based on appearance, and these inferences can affect social interactions. Although the importance of facial appearance in these judgments is well established, the impact of the body morphology remains unclear. Specifically, it is unknown whether experimentally varied body morphology has an impact on perception of threat in others. In two preregistered experiments ($N$ = 250), participants made judgments of perceived threat of body stimuli of varying morphology, both in the absence (Experiment 1) and presence (Experiment 2) of facial information. Bodies were perceived as more threatening as they increased in mass with added musculature and portliness, and less threatening as they increased in emaciation. The impact of musculature endured even in the presence of faces, although faces contributed more to the overall threat judgment. The relative contributions of the faces and bodies seemed to be driven by discordance, such that threatening faces exerted the most influence when paired with non-threatening bodies, and vice versa. This suggests that the faces and bodies were not perceived as entirely independent and separate components. Overall, these findings suggest that body morphology plays an important role in perceived threat and may bias real-world judgments.

## 1. Introduction

Physical appearance plays a central role in the impressions and inferences we draw from others. We can make judgments about characteristics such as attractiveness, likeability, competence and aggressiveness in less than 100ms [1, 2]. Unfortunately, although appearance alone is an unreliable indicator of actual personality traits [3, 4], appearance-based judgments can significantly affect social outcomes, from elections to criminal sentencing [5, 6]. Hence, appearances not only influence our perceptions of others [7], but also contribute to the biases we form about them. To understand the effects of these biases, it is therefore important to identify how and when they arise. In this study, we investigated the effect of varying body morphology on threat perception.

The influence of specific *facial* characteristics on appearance-based judgments is well documented [8, 9]. However, there is a surprisingly limited amount of research on the effect of

**Funding:** The authors received no specific funding for this work.

**Competing interests:** The authors have declared that no competing interests exist.

*body* size on perceived personality traits. This is despite evidence for the hypothesis forwarded [10] that people do not perceive others as separate body and face components. Rather, they are perceived as elements of a greater, whole-person unit [11, 12] that can encompass different properties to that of a body and face seen in isolation. In this way, the perception of a body and face in tandem may diverge from the sum of their separately perceived properties. Aviezer, Trope & Todorov [10] found evidence for their hypothesis across a number of experiments, showing that pairing emotional faces with bodies expressing an emotion incongruent with that of the face significantly impaired facial emotion identification. Indeed, body posture and morphology has been shown to influence a person's perceived emotional state [13–15], even when participants are incentivised to ignore body information [16].

Although the influence of bodies on emotion perception is established, the influence of systematic variation in body morphology on personality trait inferences is less clear. Emotion perception diverges from trait perception in that emotions are often transient and dynamic whereas (some) character traits are more stable. While emotions tend to be driven causally by specific factors, and are thus more distinct and short in nature [17], character traits are more consistent over longer periods of time [18]. Due to this distinction, the consequences of misjudging someone's emotional state as opposed to a character trait also diverge. Although someone with a clear emotional expression, such as anger, may be perceived as threatening, this perception may be isolated to instances wherein the person is judged to be angry. A subsequent bias towards or away from such an individual due to the inferred transient emotional cue may be similarly transient, while a bias based on a inferred stable trait may be similarly fixed [19]. Hence, it is vital to further our understanding of how such stable personality trait inferences, such as that of perceived threat in the absence of clear emotional cues, arise.

Ecological theory would imply that such social inferences serve an adaptive purpose [20]. For example, a slight resemblance of a neutral face to an emotionally expressive one can serve as a signal of general valence. Hence, a neutral face that resembles a happy one may serve as a signal of an amiable or approachable person. Similarly, facial masculinity and maturity can act as signals of dominance [21]. A similar phenomenon may emerge with certain cues associated with body size. Research on the role of body morphology has focused mainly on traits of leadership and dominance. Taller people tend to be perceived as more impressive, competent, social [22, 23] and are more likely to reach leadership positions [5, 24]. Indeed, more recent research [25] explored personality inferences made from computer generated body shapes of varying morphology. They found that male bodies with trim builds, wide shoulders and an inverted-triangle shaped torso tended to be associated with more dominant, extraverted personality traits.

Moreover, some work has hinted at a link between body morphology and the perception of threat and guilt [26, 27]. Here, biases often emerge against larger people. For example, Schvey et al. [28] found that, in mock trials, male jurors were more likely to judge a mugshot of an obese woman as guilty than a mugshot of a typical weighted woman. Also, Hester & Gray [29] found that, for black males, simply being tall increased perceptions of threat, with taller black males significantly more likely to be stopped by police. Finally, Palmer-Hague, Twele & Fuller [30] showed that female Ultimate Fighting Championship (UFC) fighters with higher body-mass index (BMI) are perceived as both more aggressive and more threatening. These results align with research on stigma, where overweight people are often explicitly perceived as weak-willed, lazy and undisciplined [31, 32].

While the contribution of body-information to social perception and trait inference has received less focus than that of faces, the existing work strongly indicates that body morphology does matter. However, this has yet to be systematically measured with realistic body stimuli. Moreover, the contribution of body morphology to trait inferences in the presence of

facial information remains unexplored. Until we understand the link between body morphology and trait perception, it remains difficult to test and prescribe methods to alleviate the effects of potential biases. Perceived threat is a particularly vital inference yet to be explored [25], not only due to its evolutionary importance [26], but also due to its capacity to potentially promote biases and to alter social outcomes, such as leadership contests and criminal sentencing. Experimental studies on the effects of body morphology have been primarily restricted to work still using faces as the presented stimuli [28, 30], or more abstract, unrealistic body stimuli [33, 34].

Given earlier research, we hypothesised that changes in body morphology would significantly alter the perceived threat of a body. We examined this in a preregistered online experiment, where participants rated the perceived threat of computer-generated (CG) bodies that varied systematically in level of musculature, portliness and emaciation. In a second preregistered experiment, we investigated the relative effects of varying both facial and body information on perceived threat. Given previous work outlining the holistic processing of faces and bodies [10], we hypothesised that each would play significant roles in the perception of threat.

## 2. Experiment 1

We first examined the effect of systematically varying body morphology on perceived threat. Participants were presented with a series of experimentally manipulated CG body stimuli and a subset of CG face stimuli [35] specifically designed to vary in levels of perceived threat. Our first hypothesis (H1) predicted that, in line with previous work [35] there would be strong consensus among participants' judgments on the perceived threat of the face stimuli. Second, we expected that perception of threat in the face stimuli would vary with the threat dimension value rating of the face, such that faces higher on the dimension would be perceived as more threatening (H2). Third, given the strong consensus amongst participants in previous studies on perceived threat level in faces, we expected that a strong consensus would emerge amongst participants' judgments on the perceived threat of the body stimuli (H3).

Moreover, it was expected that perceptions of threat in the body stimuli would vary as functions of variation in musculature (H4), emaciation (H5) and portliness (H6). These were non-directional hypotheses. We also expected that attributes of the participants' appearance may influence the perceived threat of the stimuli. Given research showing that people with irregular eating habits have more fearful reactions to stimuli depicting overweight people [36], we expected that participant BMI may have an effect on perceived threat (H7). Also, as taller people self-report as more assertive, dominant [37] and less socially anxious [38], we expected that taller participants would assign lower values of threat to the stimuli relative to shorter participants (H8).

Finally, as previous work has shown that more attractive faces are perceived as more trustworthy [2, 27], we recorded perceived attractiveness and expected it to have a moderating effect, such that more attractive stimuli would be rated as less threatening (H9). In addition, it has also been shown that age [39] and educational attainment [40] relate to perceived everyday danger and hostility, with older, more educated people perceiving less risk in everyday life, therefore these measures were also recorded.

### 2.1. Methodology

The experiment and all hypotheses were preregistered before data collection began (https://osf.io/bz6cg). All stimuli, data and analysis code are available at https://osf.io/7jsp9/. Ethical approval was granted by the Queen Mary University of London Institutional Review Board.

**2.1.1. Participants.** Based on previous work in this area [7], we assessed that a planned sample size of 150 participants would be more than sufficient to detect a medium-strong effect size with relatively high confidence. The experiment was conducted in February 2019 via the online software Qualtrics, with participants recruited from Prolific, an online crowdsourcing platform. To maximise the diversity of the sample, we allowed participants who were located in any part the UK, over 18 years of age, fluent English speakers and had achieved an approval rate of at least 85% in their previous Prolific study participations (63 male, 87 female; age: $M$ = 37 years, $SD$ = 12). Thirty-one individuals opted not to input their height and weight information.

**2.1.2. Stimuli (faces).** Face stimuli were taken from a dataset produced by Todorov et al. [35]. They employed a data-driven approach to estimate unbiased models of social judgments [41]. This approach identifies the information in the face that is used to make specific social judgments while imposing as few constraints as possible. In this approach, every face is considered a point in a multidimensional face space, from which any number of faces can be generated. By recording participant ratings of a particular trait, the authors create a parametrically controlled model of that trait, that accounts for a maximum amount of variance in that trait. This model is then applied to a novel face to create versions of that face that vary along the same trait dimension. For a dataset of 25 identity faces, the authors generated variations along the respective dimension: -3, -2, -1, 0, +1, +2, and +3 SD levels. They subsequently validated their dataset for each of the dimensions produced, one of which was threat. We randomly selected one of these faces that varied along this dimension of threat. This gave us seven distinct face stimuli (one for each level of SD), and each was presented facing forward with no background.

**2.1.3. Stimuli (bodies).** The stimuli were realistic CG human male figures created using state-of-the-art design software (Daz Studio 4.10 Pro: https://www.daz3d.com/daz_studio) with the Male Anatomy Smart Content package. This software provides a default, anatomically accurate model ('Genesis 8 Basic Male') with dimensions that can be precisely modified to allow fine control over individual body shapes and proportions. It is also equipped with three distinct morphological scales with which to manipulate the body systematically in a holistic fashion. The three scales are: portliness, musculature and emaciation.

We manipulated the body's appearance by applying increments on the three scales to an initial Standard body stimulus (Genesis 8 Basic Male). This Standard body is initially set to 0% on each of the three scales and incremented in steps of 16.67% across the three scales. This resulted in 6 stimuli varying in portliness, 6 stimuli varying in musculature and 6 stimuli varying in emaciation, with the Standard body stimulus representing 0% on each scale (see Fig 1). The default faces of the stimuli were blurred using Adobe Photoshop photo-editing software. The exact dimensions of each of these stimuli can be found in S1–S4 Tables.

**2.1.4. Procedure.** This experiment followed a within-participant design. All participants rated 26 different stimuli presented in a randomised order, once on threat and once on attractiveness resulting in a total of 52 trials. Upon recruitment, participants read an information sheet and then provided written informed consent. Participants completed two blocks of ratings. In one of these, participants viewed the 7 face stimuli and 19 body stimuli and were asked to rate the threat of the stimuli on a scale from 1 ("Not At All Threatening") to 7 ("Extremely Threatening"). In the other block, participants again viewed the stimuli, but this time rated their attractiveness on a scale from 1 ("Not At All Attractive") to 7 ("Extremely Attractive"). The order of presentation of the blocks was randomised across participants, such that approximately half rated the stimuli on threat first followed by attractiveness and vice versa. Each block began with an instruction screen that specified the trait to be judged (i.e. threat vs attractiveness). Participants viewed each stimulus one at a time in a randomised order. No time

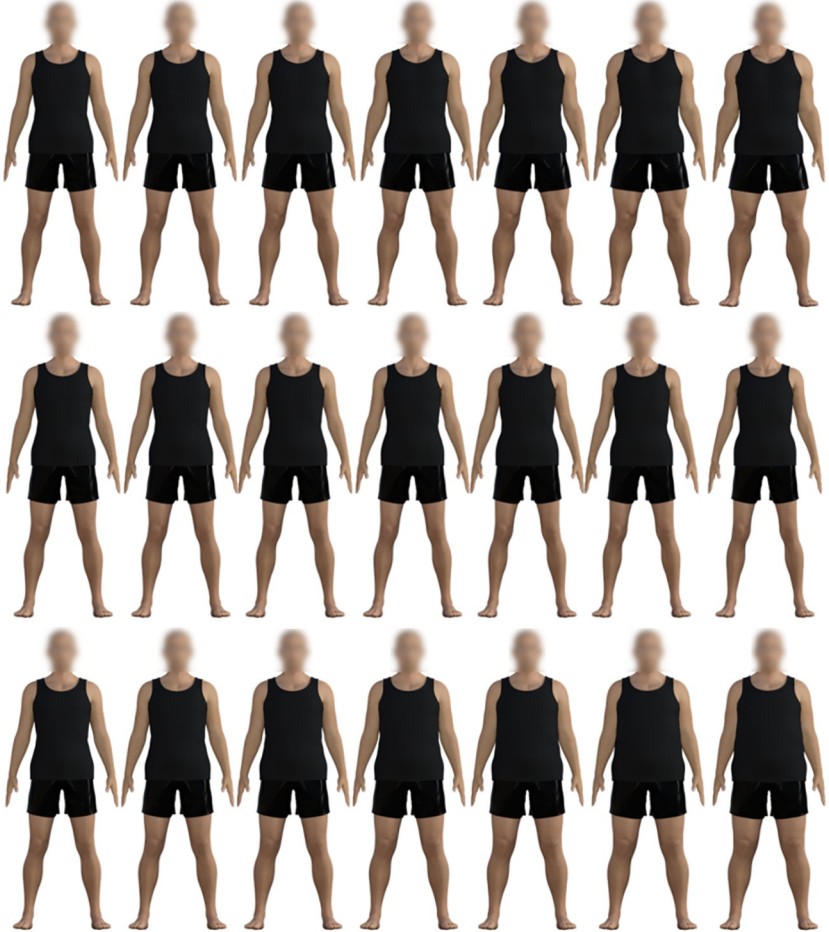

**Fig 1. Illustration of the various body morphologies presented.** Each of the three scales was applied to a standard body (far left) in increments of 16.67%. This resulted in 3 scales of stimuli varying in musculature (top), emaciation (middle) and portliness (bottom).

limit was imposed on them, and instructions encouraged them to use the full range of the response scale. After completing the two blocks of ratings, participants filled out a short demographic questionnaire, where they indicated their age, gender, height, weight and level of educational attainment (sample demographic questionnaire available at https://osf.io/kdmw8/).

**2.1.5. Data analysis.** All statistical analyses were carried out using Stata (Statistics/Data Analysis) Version 13.0. Interrater consensus was first assessed by computing intraclass correlation coefficients (*ICCs*) for perceptions of threat and attractiveness. Perceptions of threat in the face and body stimuli were then modelled. Our initial analysis plan had outlined the use of linear models (as outlined in the preregistered protocol) to investigate perceived threat. However, although the output of ordinal scales is often analysed as continuous data, given the novel nature of the current investigation, it was decided that it would be more suitable to employ a form of analysis specifically designed for ordinal data. This would provide a more apt and conservative estimate of the relationships between our predictor variables and perceived threat. Therefore, we use mixed-effects ordered logistic regressions to estimate the influence of the different independent variables on perceptions of threat, assuming a random effect across participants, with robust standard errors clustered at the participant level to account for intra-participant correlation.

Akaike Information Criterion (AIC) was used to select the best-fitting fully-powered models, the results of which are reported below. Threshold for significance (alpha) was set at a value of.05. Additional robustness checks were carried out using linear models, treating perceived threat as a continuous interval variable.

For each of the four manipulations of interest (face dimension, portliness, musculature and emaciation), we estimated separate models to investigate the effects of our predictors on perceived threat (see S5–S8 Tables for complete lists of odds ratios). For the body morphology models, the primary independent variable (IV) of interest was the total change in body morphology (in centimetres) as the bodies were manipulated along their respective dimension using Daz Studio. A breakdown of the total specific morphology changes can be found in S1–S4 Tables.

## 2.2. Results

**2.2.1. Reliability of trait-based inferences.** *ICCs* were computed according to two-way random effects models—type *ICC*(2, k) [42]. A high *ICC* indicates that the total variance in ratings is mainly explained by rating variance across stimuli instead of across participants.

In line with H1 and prior literature [35], our results showed high consensus among participants for perceptions of threat for the face stimuli, *ICC* = .99, *F*(6, 894) = 173.46, 95% *CI* = [.97, 1.00]. Also in line with H3, for the varying dimensions of the body stimuli, consensus was high for perceptions of threat in the musculature stimuli, *ICC* = .97, *F*(6, 894) = 108.02, 95% *CI* = [.93,.99]; emaciation stimuli, *ICC* = .79, *F*(6, 894) = 13.10, 95% *CI* = [.57,.95] and portliness stimuli, *ICC* = .87, *F*(6, 894) = 18.87, 95% *CI* = [.71,.97].

Relatively high consensus also emerged for ratings of attractiveness across the stimuli. For the face stimuli, *ICC* = .94, *F*(6, 894) = 27.64, 95% *CI* = [.85,.99]. For the varying dimensions of the body stimuli, consensus was high for perceptions of attractiveness in the musculature stimuli, *ICC* = .80, *F*(6, 894) = 108.02, 95% *CI* = [.57,.96]; emaciation stimuli, *ICC* = .79, *F*(6, 894) = 13.54, 95% *CI* = [.58,.95] and portliness stimuli, *ICC* = .98, *F*(6, 894) = 102.86, 95% *CI* = [.95, 1.00].

**2.2.2. Modelling perception of threat.** *2.2.2.1. Face dimension.* The overall model was significant in predicting the variation in perceived threat of the face stimuli, Wald $X^2$(8) = 211.98, *p < .001*. As predicted in H2, we found a significant stepwise, linear impact of threat dimension, with each increment accompanied by an increase in perceived threat, *OR* = 2.89, 95% *CI* = [2.48, 3.38], *p < .001*. (see Fig 2). In line with H9, a significant effect also emerged for perceived attractiveness, with more attractive stimuli perceived as less threatening, *OR* = 0.81, 95% *CI* = [0.66, 0.98], *p* = .03. The model also revealed a significant effect (*p* = .01) of

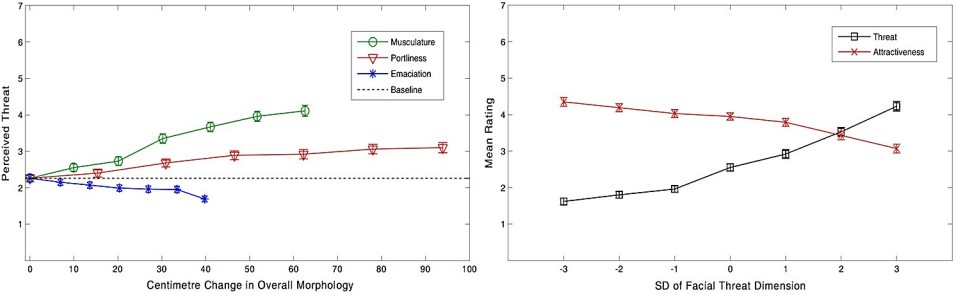

**Fig 2. Mean perceived threat for each of the levels of body morphology manipulations by total centimetre change in morphology (left) and mean perceived threat and perceived attractiveness for each of the levels of facial dimension (right).** Error bars represent standard errors.

participant age on perceived threat, such that older participants tended to view the stimuli as less threatening compared to younger participants. Specifically, participants aged 18–30, $OR$ = 7.81, 95% $CI$ = [2.38, 25.61], $p >$.001 and aged 30–45, $OR$ = 5.06, 95% $CI$ = [1.53, 16.69], $p$ = .008 found the stimuli significantly more threatening than those aged 46–60. Finally, a significant effect of education emerged, such that participants with undergraduate degrees, $OR$ = 3.31, 95% $CI$ = [1.35, 8.09], $p$ = .009, master's degrees, $OR$ = 4.01, 95% $CI$ = [1.31, 12.26], $p$ = .02 and doctorate degrees, $OR$ = 3.67, 95% $CI$ = [1.32, 10.18], $p$ = .01 rated the stimuli as more threatening than those without degrees. Contrary to H7 and H8, the assessment of height and BMI revealed no significant effects.

Robustness checks using linear models indicated that facial dimension accounted for a large proportion of the variance in perceived threat; $F(1, 149)$ = 337.22, $p < .001$, $R^2$ = .31, with overall variance explained increasing with the addition of perceived attractiveness, age, gender, education and order of presentation; $F(8, 149)$ = 47.23, $p < .001$, $R^2$ = .37.

*2.2.2.2. Musculature*. For musculature, the overall model was significant in predicting the variation in perceived threat of the stimuli, Wald $X^2(9)$ = 180.90, $p < .001$. Consistent with H4, we found that the level of musculature of the body stimuli had a significant positive relationship with the threat perceived in the body, $OR$ = 1.08, 95% $CI$ = [1.07, 1.10], $p < .001$. In other words, the more muscular the stimulus, the more threatening it appeared (see Fig 2). With respect to the 7 levels of musculature seen by participants, a one unit increase in level of musculature saw the odds of rating the stimulus as more threatening increase by a factor of 2.33, 95% $CI$ = [2.05, 2.64], $p < .001$.

The model also revealed a significant effect ($p$ = .002) of participant age on perceived threat, such that older participants tended to view the stimuli as less threatening compared to younger participants. Specifically, participants aged 18–30, $OR$ = 29.13, 95% $CI$ = [6.27, 135.30], p $>$.001 and aged 30–45, $OR$ = 9.61, 95% $CI$ = [1.73, 53.28], $p$ = .01 found the stimuli significantly more threatening than those aged 46–60. Contrary to H7 and H8, the assessment of height and BMI revealed no significant effects. Although perceived attractiveness and gender were not seen to have significant main effects, further analysis revealed a significant interaction effect ($p$ = .01). This showed that, for male participants, there was a significant relationship between perceived attractiveness and perceived threat, such that more attractive bodies were perceived as more threatening, $OR$ = 1.50, 95% $CI$ = [1.11, 2.03], $p$ = .008.

Robustness checks using linear models indicated that musculature accounted for a significant amount of the variance in perceived threat; $F(1, 149)$ = 252.60, $p < .001$, $R^2$ = .16, with overall variance explained increasing with the addition of perceived attractiveness, age, gender, education, interaction of perceived attractiveness and gender, and order of presentation; $F(9, 149)$ = 36.69, $p < .001$, $R^2$ = .22.

*2.2.2.3. Emaciation*. For emaciation, the model was again significant in predicting the variation in perceived threat of the stimuli, Wald $X^2(8)$ = 90.68, p $< .001$. Consistent with H5, we found that the level of emaciation of the body stimuli had a significant negative relationship with the threat perceived in the body, OR = 0.96, 95% CI = [0.94, 0.97], p $< .001$ (see Fig 2). With respect to the 7 levels of emaciation seen by participants, a one unit increase in level of emaciation saw the odds of rating the stimulus as threatening decrease by a factor of 0.74, 95% CI = [0.67, 0.80], p $< .001$.

The model also revealed a significant effect ($p$ = .009) of participant age on perceived threat, such that older participants tended to view the stimuli as less threatening compared to younger participants. Specifically, participants aged 30–45, $OR$ = 0.14, 95% $CI$ = [0.05, 0.42], $p < .001$ and aged 31–46, $OR$ = 0.04, 95% $CI$ = [0.01, 0.17], $p < .001$ found the stimuli significantly less threatening than those aged 18–30. Finally, a significant effect of education emerged, where participants with undergraduate degrees, $OR$ = 10.50, 95% $CI$ = [3.23, 34.18], $p < .001$ and

master's degrees, $OR = 5.56$, 95% $CI = [1.21, 25.49]$, $p = .02$, found the stimuli significantly more threatening than those with no degree. Contrary to H7, H8 and H9, the assessment of perceived attractiveness, height and BMI revealed no significant effects.

Robustness checks using linear models indicated that emaciation accounted for a small, but significant amount of the variance in perceived threat; $F(1, 149) = 46.19$, $p < .001$, $R^2 = .02$, with overall variance explained increasing with the addition of perceived attractiveness, age, gender, education, and order of presentation; $F(8, 149) = 9.50$, $p < .001$, $R^2 = .12$.

*2.2.2.4. Portliness.* For portliness, the model was significant in predicting the variation in perceived threat of the stimuli, Wald $X^2(8) = 43.50$, $p < .001$. Consistent with H6, we found that the level of portliness of the body stimuli had a significant positive relationship with how threatening it appeared, $OR = 1.02$, 95% $CI = [1.01, 1.02]$, $p < .001$ (see Fig 2). With respect to the 7 levels of portliness seen by participants, a one unit increase in level of portliness saw the odds of rating the stimulus as more threatening increase by a factor of 1.30, 95% $CI = [1.17, 1.44]$, $p < .001$.

The model also revealed a significant effect ($p = .02$) of participant age on perceived threat, such that older participants tended to view the stimuli as less threatening compared to younger participants. Specifically, participants aged 31–45, $OR = 0.26$, 95% $CI = [0.10, 0.67]$, $p = .005$ and aged 46–60, $OR = 0.09$, 95% $CI = [0.02, 0.35]$, $p < .001$ found the stimuli significantly less threatening than those aged 18–30. In addition, a significant effect of education emerged, where participants with undergraduate degrees found the stimuli significantly more threatening than those without a degree, $OR = 3.73$, 95% $CI = [1.22, 11.43]$, $p = .02$. Contrary to H7, H8 and H9, the assessment of perceived attractiveness, height and BMI revealed no significant effects.

Robustness checks using linear models indicated that portliness accounted for a small, but significant amount of the variance in perceived threat; $F(1, 149) = 39.23$, $p < .001$, $R^2 = .04$, with overall variance explained increasing with the addition of perceived attractiveness, age, gender, education, and order of presentation; $F(8, 149) = 6.48$, $p < .001$, $R^2 = .10$.

## 3. Experiment 2

Experiment 1 showed that perceived threat can shift with systematic changes in body morphology. To assess whether this persists in the presence of facial information, and to explore the interaction of faces and bodies in the perception of threat, Experiment 2 examined perceived threat in a series of face+body compound stimuli. We presented participants with stimuli that consisted of combinations of faces of varying threat dimension with bodies of varying levels of musculature. Musculature was chosen as this manipulation produced the strongest effect in Experiment 1. Participants were also presented with the face-only and body-only stimuli independently. In line with Experiment 1, we predicted strong consensus on the perceived threat of the stimuli (H1). We expected that perceived threat would significantly vary with changes in facial dimension for the face-only stimuli (H2) and with changes in musculature for the body-only stimuli (H3). For the compound stimuli, we predicted that perceived threat would increase with both facial dimension (H4) and musculature (H5). We expected perceived attractiveness would have a negative relationship with perceived threat for the face and compound stimuli (H6).

Furthermore, work in emotion-recognition [15, 43] shows that, when facial cues are ambiguous, people rely more heavily on body cues when ascribing specific emotions to full-body stimuli. Hence, we hypothesised that perceived threat in the compounds would be most strongly correlated with the ratings given to the body-only stimuli when the threat dimension of the face stimuli was ambiguous, i.e., of a neutral level not clearly signalling either presence or

absence of threat (H7), and with the ratings given to the face-only stimuli when the morphology of the body stimuli was relatively neutral (H8). Finally, we predicted that perceived threat in the compound stimuli would be primarily driven by the face when the threat dimension of the face was unambiguous (clearly threatening or non-threatening) and driven by the body when the threat dimension of the face was ambiguous (H9).

## 3.1. Methodology

The experiment and all hypotheses were preregistered before data collection began (https://osf.io/xhtr9). All stimuli, data and analysis code are available at https://osf.io/s97ka/. Ethical approval was granted by the Queen Mary University of London Institutional Review Board.

**3.1.1. Participants.** Using the data from Experiment 1, we reran the analyses with increasingly smaller random samples to determine the minimum required sample size for Experiment 2. It was determined that a sample size of 100 participants would be more than sufficient to replicate the main effects.

The experiment was conducted in June 2019 via the online platform Qualtrics, with participants recruited via the Prolific participant body. To maximise the diversity of the sample, we allowed participants who were located in any part of the UK, over 18 years of age, fluent English speakers, had achieved an approval rate of at least 85% in their previous Prolific study participations and had not participated in Experiment 1 (26 male, 74 female; age: M = 38 years, $SD$ = 12).

**3.1.2. Stimuli (faces).** We selected two face identity stimuli from the same dataset as in Experiment 1 [35]. Given 7 levels of SD, this produced 14 distinct face stimuli, half of which were used to create the compound stimuli (see below), with the other half presented independently as additional face-only distractor stimuli. These distractors were included to decrease experimenter demand, such that the manipulations of the compound stimuli were less apparent: not all presented faces were also related to a compound stimulus. In addition, they made it more difficult for the participants to recall a face they had previously been shown in isolation on a compound, and simply rate the compound with the same threat previously given to just the face. In order to create realistic compound stimuli, we converted the faces to greyscale and adjusted the lightness of the skin tone to match that of the bodies.

**3.1.3. Stimuli (bodies).** Due to the pronounced effect of musculature on perceived threat in Experiment 1, we used the musculature-varying body stimuli only in Experiment 2. As in Experiment 1, we had 7 body stimuli that varied systematically in overall musculature. To combine these bodies with the face stimuli such that the compounds appeared natural, we converted the bodies to greyscale.

**3.1.4. Stimuli (compounds).** Using Adobe Photoshop, we created grey scale compound stimuli by combining the face and body stimuli. The sizes of these face stimuli were adjusted to combine more believably with their respective body combinations. These adjustments were very slight, with the maximum disparity between the smallest and largest versions of an individual head at about 7%. While this may have introduced some slight variability in how the faces were perceived, it was decided that this was preferable to presenting noticeably incongruent face and body combinations. Each of the 7 experimental face stimuli was combined with each of the 7 body stimuli. This resulted in a total of 49 compound stimuli.

**3.1.5. Procedure.** The experiment was conducted in a similar fashion to Experiment 1. Participants completed two blocks of ratings, consisting of 35 trials each. In each block they were shown half of the stimuli in a randomized order with a break between blocks. Participants completed 70 trials in total, comprised of the 7 experimental face stimuli, the 7 distractor face stimuli, the 7 body-only stimuli and the 49 compound stimuli. They rated, on separate 7-point

scales, how threatening and attractive they found each stimulus. On completion, participants were asked to fill out a short demographics questionnaire before exiting the experiment (sample demographic questionnaire available at https://osf.io/4bju3/).

**3.1.6. Data analysis.** Interrater consensus was again first assessed by computing *ICC*s for perceptions of threat and attractiveness. Perceptions of threat in the face, body and compound stimuli were then modelled as in Experiment 1 (see S9–S12 Tables for complete lists of odds ratios). Akaike Information Criterion (AIC) was used to select the best-fitting fully-powered models, the results of which are reported below. Finally, as an exploratory analysis, we examined the interaction between facial and body information in the compound stimuli by plotting the differences between the threat ratings given to the compound stimuli and the threat ratings given independently to their separate corresponding face and body components.

## 3.2. Results

**3.2.1. Reliability of trait-based inferences.** In line with H1, we found high consensus among participants for perceptions of threat for the experimental face stimuli, *ICC* = .98, *F*(6, 594) = 125.24, 95% *CI* = [.96, 1.00], distractor face stimuli, *ICC* = .98, *F*(6, 594) = 81.47, 95% *CI* = [.94,.99], body-only stimuli, *ICC* = .91, *F*(6, 594) = 33.77, 95% *CI* = [.81,.98] and compound stimuli, *ICC* = .96, *F*(48, 4752) = 57.57, 95% *CI* = [.94,.98]. High consensus also emerged for ratings of attractiveness for the experimental face stimuli, *ICC* = .93, *F*(6, 594) = 24.69, 95% *CI* = [.83,.98] and distractor faces, *ICC* = .82, *F*(6, 594) = 11.03, 95% *CI* = [.62,.96]. Lower attractiveness consensus emerged for the body-only stimuli relative to Experiment 1, *ICC* = .49, *F*(6, 594) = 3.88, 95% *CI* = [.83,.98] and compound stimuli, *ICC* = .35, *F*(48, 4752) = 2.17, 95% *CI* = [.18,.53]. This decrement in attractiveness consensus for the body stimuli may be attributable to their presentation in the same block as the compound stimuli, wherein the presence of contrasting face/body combinations may have influenced participants' expectations of the reliability of the bodies' cue signals.

**3.2.2. Modelling perception of threat.** *3.2.2.1. Face-only stimuli.* The overall models were significant in predicting the variation in perceived threat of both the experimental face-only stimuli (Wald $X^2$(8) = 235.36, *p* < *.001*) and the distractor face-only stimuli (Wald $X^2$(8) = 181.47, *p* < *.001*). In line with H2, we found that the threat dimension of faces had a significant positive relationship with perceived threat in both the experimental face stimuli, *OR* = 2.56, 95% *CI* = [2.24, 2.93], *p* < *.001* and the distractor face stimuli, *OR* = 2.36, 95% *CI* = [2.05, 2.70], *p* < *.001*. Also, in line with H6, a significant negative effect of perceived attractiveness emerged for both experimental faces, *OR* = 0.65, 95% *CI* = [0.52, 0.80], *p* < *.001* and distractor faces, *OR* = 0.71, 95% *CI* = [0.52, 0.95], *p* = .02. Robustness checks using linear models indicated that facial dimension accounted for a large proportion of the variance in perceived threat; *F*(1, 99) = 337.22, *p* < *.001*, $R^2$ = .33, with overall variance explained increasing with the addition of perceived attractiveness, age, gender, education and order of presentation; *F*(8, 99) = 35.86, *p* < .001, $R^2$ = .35.

*3.2.2.2. Body-only stimuli.* For the body-only stimuli, the model was significant in predicting the variation in perceived threat of the stimuli, Wald $X^2$(1) = 82.63, *p* < *.001*. Consistent with Experiment 1, in support of H3, we found that the level of musculature of the body stimuli had a significant positive relationship with the threat perceived in the body, *OR* = 1.05, 95% *CI* = [1.04, 1.06], *p* < *.001*. In other words, the more muscular the stimulus, the more threatening it appeared. With respect to the 7 levels of musculature seen by participants, a one unit increase in level of musculature saw the odds of rating the stimulus as more threatening increase by a factor of 1.70, 95% *CI* = [1.52, 1.91], *p* < *.001*. AIC [44] indicated that the most simple model (assessing only the effects of musculature on threat) was the best fitting, with none of the other

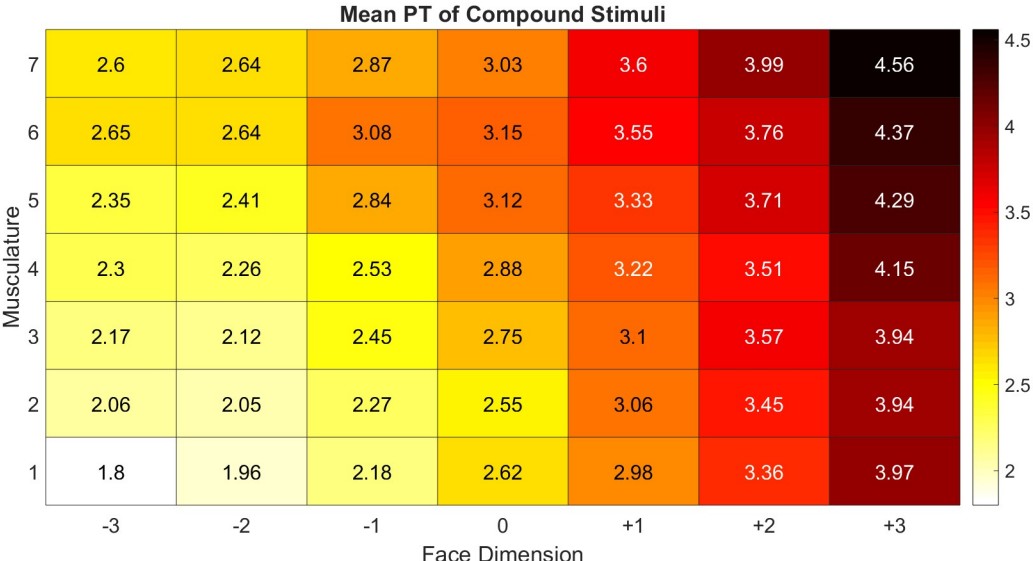

**Fig 3. Heatmap of mean perceived threat (PT) (on a scale of 1–7) for the compound stimuli.** Compounds increase in facial dimension along the X axis. Compounds increase in musculature along the Y axis.

predictors significantly predicting threat or improving model fit. Robustness checks using linear models indicated that musculature accounted for a significant amount of the variance in perceived threat; $F(1, 99) = 83.39$, $p < .001$, $R^2 = .09$.

*3.2.2.3. Compound stimuli.* For the compound stimuli, the model was significant in predicting the variation in perceived threat of the stimuli, Wald $X^2(11) = 444.68$, $p < .001$. Although perceived attractiveness did not have a significant main effect, further analysis revealed a significant interaction effect ($p = .006$) between perceived attractiveness and gender. This showed that, for male participants, there was a significant relationship between perceived attractiveness and perceived threat, such that more attractive stimuli were perceived as less threatening, $OR = 0.38$, 95% $CI = [0.21, 0.69]$, $p = .001$, lending some support for H6.

The model revealed significant main effects of both level of musculature of the compound ($OR = 1.04$, 95% $CI = [1.03, 1.04]$, $p < .001$) and facial dimension of the compound ($OR = 2.11$, 95% $CI = [1.93, 2.32]$, $p < .001$) on threat perceived in the compound. The effects of facial dimension and musculature can be seen in the heatmap of Fig 3. The model also found a very weak, but significant ($b = -.003$, $p < .001$) interaction between face dimension and musculature on perceived threat. To assess this interaction, we estimated separate models assessing the effect of facial dimension at the 7 different levels of musculature. Similarly, we estimated separate models assessing the effect of musculature level at the 7 different levels of facial dimension (see S13 Table for a summary of odds ratios at each level). Each cut of the model found musculature and facial dimension significant ($p < .001$) in predicting perceived threat, with facial dimension consistently exhibiting a stronger effect at all levels. Robustness checks using linear models indicated that the predictors accounted for a significant proportion of the variance in perceived threat; $F(13, 99) = 27.92$, $p < .001$, $R^2 = .24$.

Although facial dimension had a stronger effect on perceived threat than musculature, we cannot claim that facial content is more important than body morphology for the perception of threat. The stronger effect may be driven by a broader range of the perceived threat in the face stimuli than in the body stimuli. Hence, as an exploratory analysis, we estimated a further model of perceived threat in the compounds, this time with perceived threat of the face-only

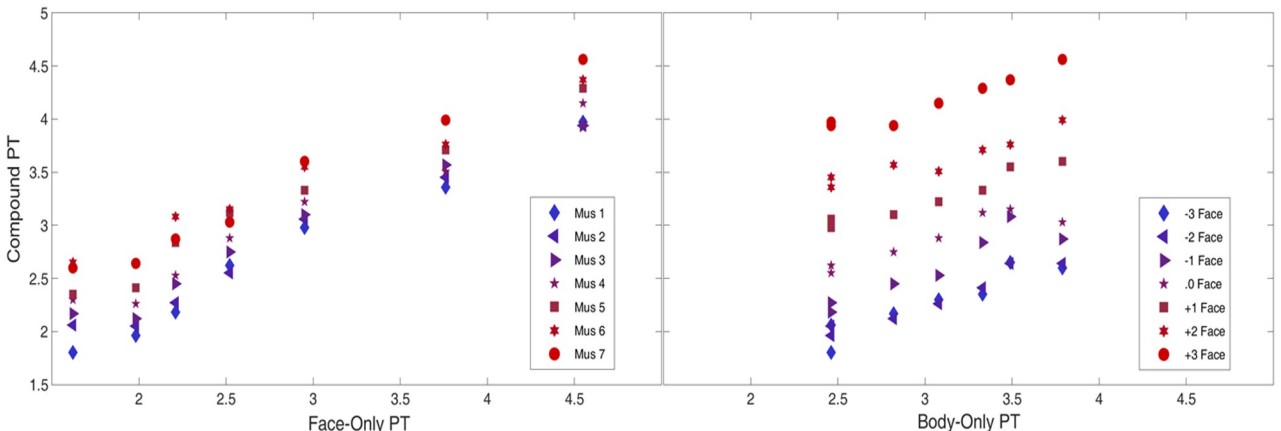

**Fig 4. Scatterplots outlining the relationship between the perceived threat (PT) in the face-only stimuli and the compound stimuli (left) and between the perceived threat in the body-only stimuli and the compound stimuli (right).** Each vertical cluster of data point corresponds to perceived threat of one level of face-only stimuli (left) or body-only stimuli (right).

and body-only stimuli acting as predictors instead of facial dimension and musculature. Here, the scales presented to participants for rating the face and body stimuli were identical (as opposed to the musculature and face scales which are different), thus allowing a more apt comparison of the effect sizes of the two predictors. Again, the overall model was significant (Wald $X^2$(10) = 356.38, $p < .001$), with both perceived threat for the face and body stimuli showing significant main effects ($p < .001$). For a one unit increase in how threatening the body was perceived, the odds of the compound being perceived as more threatening increased by a factor of 1.66. However, for a one unit increase in how threatening the face was perceived, the odds of the compound being perceived as more threatening increased by a factor of 2.80, illustrating the stronger effect of face information on perceived threat in the Compounds. These relationships are illustrated in Fig 4.

The closer relationship of facial perceived threat to compound perceived threat is also reflected in the correlation analyses. Here, a significant correlation emerged between body-only and compound stimuli ($r_s$(98) = .67, $p < .001$), while a stronger association emerged between face-only and compound stimuli ($r_s$(98) = .84, $p < .001$). Contrary to H7 and H8, although perceived threat in the compounds was significantly correlated with all levels of face-only and body-only threat, no clear pattern in correlation by level of musculature or facial dimension emerged.

**3.2.3. Analysis of disparity between compound threat and face/body threat.** As a final piece of exploratory analysis, we further explored the interaction between facial and body threat in the perceived threat of the compound stimuli. We first estimated the relative contributions of face and body morphology to perceived threat in the compound stimuli by calculating the absolute difference between the mean threat of each compound stimulus and the threat of its corresponding face-only or body-only stimuli. This was taken as an index of the extent to which the threat of the compounds deviated from threat derived solely from the faces and bodies on their own. A Wilcoxon signed-rank test showed that this absolute difference was significantly larger for the body morphology (Median = 0.55) than for face dimension (Median = 0.32), $z = 3.71$, $p < .001$.

To assess the interaction, we then computed a set of difference scores for each compound. In order to capture directionality of perceived threat (increasing or decreasing it), we kept the signed difference, where the sign (+/-) indicates whether the compounds or face/bodies were

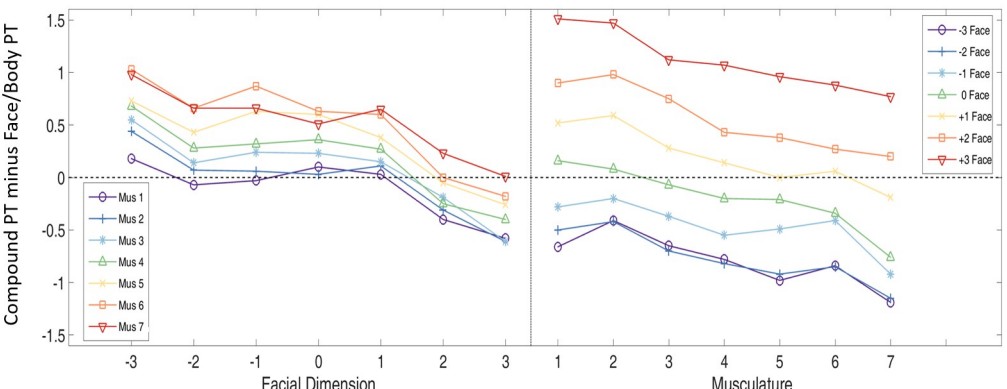

**Fig 5. Plots illustrating the disparity in perceived threat (PT) for compound stimuli minus the face-only stimuli (left) and the disparity in perceived threat for the compound stimuli minus the body-only stimuli (right).** In all conditions, the slopes of the best fitting line to the data were significantly different from 0 ($p < .05$), indicating the significant interaction of face and body information.

perceived as more threatening. The closer this score is to 0, the more closely matched the ratings were between the individual component threat (e.g. face or body) and the compound threat, whereas the further from 0, the more the threat from the two types of stimuli deviated. This distance from 0 can be conceptualised as the leftover contribution of the body and face threat to compound judgment of threat in the absence of the face and body respectively.

Plotting these out in Fig 5 illustrates how these contributions vary by level of facial dimension and musculature. The lines for facial threat subtracted from compound threat are clustered considerably more closely around 0 than those for body threat subtracted from compound threat, reflecting the higher contribution of facial information to the compound judgment. Contrary to H9, the contribution of the bodies (Compound threat minus Face-Only threat, left-hand panel) was at its greatest (greatest distance from 0) when facial threat dimension was at its lowest (-3 SD Facial Threat Dimension), and also when musculature was at its highest (Muscle Level 7 in red). Moreover, the contribution of the faces (Compound threat minus Body-Only threat, right-hand panel) was at its greatest when musculature was at its lowest (Body Musculature level 1) and facial threat dimension at its highest (+3 SD Facial Threat in red).

In the absence of any interaction between faces and bodies, the slopes of the lines would be 0, indicating that the contribution of facial dimension and musculature to perceived threat in the compounds is independent of either musculature level or facial dimension, respectively. However, in both panels, the contribution of faces and bodies seems to follow a pattern according to the discordance between the threat signal of the faces and bodies. For example, faces exercise their greatest influence (greatest distance from 0) when the most threatening face (+3 SD) is paired with the least threatening body (Level 1 Muscle). However, the influence of the most threatening face decreases (approaches 0) as it is paired with more muscular bodies. In other words, as a threatening face/body is paired with more threatening bodies/faces, the discordance between the two threat signals decreases and the independent contribution of the face/body is reduced.

## 4. Discussion

In two preregistered studies, we found evidence supporting the hypothesis that systematic changes in body morphology can significantly influence how threatening a person appears.

Judgment of threat was primarily driven by facial information, with the odds of perceiving a person as more threatening increasing nearly threefold with each unit increase in perceived facial threat. However, larger bodies also tended to be seen as more threatening than smaller bodies, both in the absence and presence of facial information. Indeed, the odds of perceiving a person as more threatening increased more than one and a half-fold with each unit increase in perceived body threat. While the association between body size and perceived negative traits is not novel, this represents the first study, to our knowledge, to demonstrate that perceived threat can shift significantly with systematic changes in body morphology. Using this methodology, we were able to directly measure the effects of body morphology on perceived threat. In Experiment 1, bodies were perceived as more threatening the larger they became, most notably with increased musculature. This finding was replicated in Experiment 2.

Our findings are consistent with Palmer-Hague, Twele & Fuller [30], who found that perceived threat in facial stimuli was significantly predicted by BMI. They are also somewhat in line with Hu et al. [25], who found that more muscular builds tend to be seen as more dominant. More generally, these results dovetail with the growing literature on the capacity of appearance to significantly affect character trait inferences, while also adding to the sizeable obesity stigma literature. It appears that larger people may be perceived as more threatening. This would make sense from an ecological theory perspective, with size potentially serving as an inferred cue of strength. This increased perceived threat may contribute to biases against larger people [28, 29]. For example, in the realm of courtroom decision-making, it has been shown that defendants who appear untrustworthy are more likely to fall victim to harsher sentencing [6]. It is conceivable that people who appear threatening may also be more severely judged.

The study also contributes to the literature on the joint processing of bodies and faces. In line with the emotion recognition literature, we found that two stimuli sharing the same facial information can be perceived as significantly different depending on body information. This common interaction of face and body information in both this study and previous work on emotion recognition is perhaps unsurprising given the link between emotions and trait perception. The perception of emotional expressions has been shown to fuel, and can directly contribute to, overgeneralisations about other people's trait characteristics [45–47]. Indeed, work by Montepare & Dobish [48] showed that actors posed with angry emotional expressions were perceived to be high in trait dominance and low in trait affiliation, while actors posing with surprise and happiness were seen as high in both trait dominance and affiliation.

However, the current study diverges from findings in emotion perception in the nature of the observed interaction of the face and body information. In contrast with work on combined emotional faces and bodies stimuli [15, 43], the contribution of the body here was maximised when paired with faces of low threat signal, rather than ambiguous threat signal. This could be attributable to the more transient nature of emotions in comparison to more stable character traits. Emotions are short and distinct feelings, which tend to have a specific cause [17], while character traits tend to be consistent over many years [18]. Similarly, the perceived emotion of a face can be rather malleable, and highly dependent on contextual and body cues [13]. Hence, contextual cues may be of particular importance when the facial cue is ambiguous. However, a face that signals a "neutral" level of a character trait such as threat may not be ambiguous or uninterpretable. Rather, it may be signalling a "medium" level of threat, an amount that can be processed and interpreted.

These results suggest that, rather than simply summing the independent threat level of the face and body, the two are integrated into a single judgment, that tends to be more heavily driven by the face. In this way, the perception of the compounds diverged from the mere sum of their separately perceived properties. The relative contributions of face and body seem to be

driven by discordance, with faces exercising their greatest influence when paired with discordant bodies, and vice versa. This may be attributable to a pop-out effect, in that faces that may not appear to "match" the accompanying body (and vice versa) may be more likely to capture attention, and thus more strongly drive the judgment of the overall compound [49]. Although not providing direct evidence for holistic processing per se, this significant interaction lends some support to the hypothesis forwarded by Aviezer, Trope & Todorov [10]; that people do not perceive others as separate body and face components. Rather, it seems likely they are perceived as elements of a greater, whole-person unit. In this case, the signals of threat from face and body are integrated such that their respective strengths are dependent on the nature of their paired signal. The holistic person-perception hypothesis has found rather consistent support, from the emotion/identity identification literature [11] to findings on gaze detection [12]. However, this study represents the first evidence for such complimentary face and body processing in the area of trait/character inference.

While we found strong evidence for our primary hypotheses, we found no effect of participant height or BMI on perceived threat. This could be attributed to the manner in which the stimuli were presented. Participants were presented with an image on screen, as opposed to judging a real-sized potential threat. In a more realistic environment, it may be that people judge potential threats in terms of the personal threat posed. In this sense, a large person may feel less threatened by a person of average build than would a small person. Here, the potential effect of relative size may have been nullified. In addition, we found that the relation between perceived attractiveness and perceived threat was somewhat inconsistent. Although more attractive faces were perceived as less threatening, males found more attractive muscular bodies to be more threatening in Experiment 1. This may be due to the male participants not finding the bodies attractive in a romantic sense, but rather in recognition of a typically attractive male form [50]. In this case, the more muscular bodies were perceived as more attractive, but did not detract from the signalled threat. However, as the current study did not record the sexual orientation of the participants, this interpretation is somewhat speculative. Future studies investigating perceived attractiveness and threat should record the sexual orientation of participants to elucidate more clearly the nature of this interaction.

In addition to our primary hypotheses, we also observed significant effects of age and education in our first experiment. Older participants tended to perceive less threat in the stimuli, which is in line with previous work [39]. Contrary to expectations, it was also noted that participants with third-level degrees tended to perceive more threat than those without a degree, which is contrary to previous indications that participants of lower educational status show more hostile reactivity to ambiguous social scenarios [40]. However, it has also been shown that those of lower social rank and education may be more adept at tracking hostility [40]. As our stimuli did not overtly indicate hostility, these participants may have thus ascribed lower threat ratings.

A number of limitations of the current study should be mentioned. First, our study was limited to body stimuli which consisted entirely of images of white males. In order to generalise these findings, it would be useful to replicate the study using both female and male stimuli. It would be particularly relevant to repeat this with stimuli of varying races given the documented bias of young black men being perceived as bigger and more physically threatening than white men [51, 52]. Furthermore, our stimuli were entirely CG. While this lent us a level of control over body morphology that would have been impossible with images of real people, it limits the ecological validity of our findings. Future studies could attempt to use photo-editing software to systematically vary the body morphology of images of real people. Finally, the stimuli presented in this study were relatively small, displayed on a computer screen. A study utilising virtual reality (VR) apparatus [53] to display life-sized human stimuli to participants,

while manipulating facial information and body morphology, may tap into a more ecological measurement of perceived threat. Furthermore, a VR study could also manipulate the participants' own virtual height, thus exploring the impact of discrepant size on perceptions of threat.

## 5. Conclusion

Here, we demonstrated that perception of threat can shift significantly with variations in body morphology. This trait's vulnerability to change with variations in physical characteristics could have far-reaching ramifications, from affecting electoral outcomes to criminal sentencing decisions. This also adds to the expanding literature on combined face and body processing, supporting the idea of complimentary processing and interaction between various signals. Limitations notwithstanding, these findings shine a light on the potential of body morphology to unfairly bias our perception of others.

## Supporting information

**S1 Table. Breakdown of the dimensions of the Daz human male body stimuli (in centimetres), varying by 7 levels of musculature.**
(DOCX)

**S2 Table. Breakdown of the dimensions of the Daz human male body stimuli (in centimetres), varying by 7 levels of emaciation.**
(DOCX)

**S3 Table. Breakdown of the dimensions of the Daz human male body stimuli (in centimetres), varying by 7 levels of portliness.**
(DOCX)

**S4 Table. Breakdown of the total change in centimetres of the Daz human male body stimuli having undergone transformations of musculature, emaciation and portliness.**
(DOCX)

**S5 Table. Odds ratios from ordered logit models predicting perceived threat in the face stimuli (Experiment 1).** Model 1 regresses PT onto face dimension only. Model 2 regresses PT onto face dimension, PA, age, gender, education and order of block presentation. Model 3 additionally includes height and BMI. The best-fitting full-powered model (Model 2) was identified using the Akaike Information Criterion (AIC), the results of which were reported in the main paper.
(DOCX)

**S6 Table. Odds ratios from ordered logit models predicting perceived threat in the musculature-varying body stimuli (Experiment 1).** Model 1 regresses PT onto change in musculature only. Model 2 regresses PT onto change in musculature, PA, age, gender, education and order of block presentation. Model 3 includes an interaction term between PA and Gender. Model 4 additionally includes height and BMI. The best-fitting full-powered model (Model 3) was identified using the Akaike Information Criterion (AIC), the results of which were reported in the main paper.
(DOCX)

**S7 Table. Odds ratios from ordered logit models predicting perceived threat in the emaciation-varying body stimuli.** Model 1 regresses PT onto change in emaciation only. Model 2 regresses PT onto change in emaciation, PA, age, gender, education and order of block presentation. Model 3 additionally includes height and BMI. The best-fitting full-powered model

(Model 2) was identified using the Akaike Information Criterion (AIC), the results of which were reported in the main paper.
(DOCX)

**S8 Table. Odds ratios from ordered logit models predicting perceived threat in the portliness-varying body stimuli.** Model 1 regresses PT onto change in portliness only. Model 2 regresses PT onto change in portliness, PA, age, gender, education and order of block presentation. Model 3 additionally includes height and BMI. The best-fitting full-powered model (Model 2) was identified using the Akaike Information Criterion (AIC), the results of which were reported in the main paper.
(DOCX)

**S9 Table. Odds ratios from ordered logit models predicting perceived threat in the experimental face stimuli (Experiment 2).** Model 1 regresses PT onto face dimension only. Model 2 regresses PT onto face dimension, PA, age, gender, education and order of block presentation. Model 3 additionally includes height and weight. The best-fitting model (Model 2) was identified using the Akaike Information Criterion (AIC), the results of which were reported in the main paper.
(DOCX)

**S10 Table. Odds ratios from ordered logit models predicting perceived threat in the distractor face stimuli (Experiment 2).** Model 1 regresses PT onto face dimension only. Model 2 regresses PT onto face dimension, PA, age, gender, education and order of block presentation. Model 3 additionally includes height and weight. The best-fitting model (Model 2) was identified using the Akaike Information Criterion (AIC), the results of which were reported in the main paper.
(DOCX)

**S11 Table. Odds ratios from ordered logit models predicting perceived threat in the musculature-varying body stimuli (Experiment 2).** Model 1 regresses PT onto change in musculature only. Model 2 regresses PT onto change in portliness, PA, age, gender, education and order of block presentation. Model 3 additionally includes height and weight. The best-fitting model (Model 1) was identified using the Akaike Information Criterion (AIC), the results of which were reported in the main paper.
(DOCX)

**S12 Table. Odds ratios from ordered logit models predicting perceived threat in the compound body stimuli.** Model 1 regresses PT onto face dimension and change in musculature only. Model 2 regresses PT onto face dimension, change in musculature, PA, age, gender, education and order of block presentation. Model 3 includes an interaction term between face dimension and change in musculature, and PA and Gender. Model 4 additionally includes height and weight. The best-fitting interaction model (Model 3) was identified using the Akaike Information Criterion (AIC), the results of which were reported in the main paper.
(DOCX)

**S13 Table. Odds ratios indicating the effects of facial threat and musculature on compound perceived threat at the varying cuts of musculature and facial threat respectively.**
(DOCX)

## Acknowledgments

We thank Robert Lachlan and Shane Timmons for helpful discussions.

## Author Contributions

**Conceptualization:** Terence J. McElvaney, Magda Osman, Isabelle Mareschal.

**Data curation:** Terence J. McElvaney.

**Formal analysis:** Terence J. McElvaney, Isabelle Mareschal.

**Investigation:** Terence J. McElvaney.

**Methodology:** Terence J. McElvaney.

**Project administration:** Isabelle Mareschal.

**Software:** Terence J. McElvaney.

**Supervision:** Isabelle Mareschal.

**Visualization:** Terence J. McElvaney.

**Writing – original draft:** Terence J. McElvaney.

**Writing – review & editing:** Terence J. McElvaney, Magda Osman, Isabelle Mareschal.

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
