## [Decision Letter · Decision Letter 0]

4 Nov 2020

PONE-D-20-19322

Perceiving threat in others: the role of body morphology

PLOS ONE

Dear Dr. McElvaney,

Thank you for submitting your manuscript to PLOS ONE. After careful consideration, we feel that it has merit but does not fully meet PLOS ONE’s publication criteria as it currently stands. Therefore, we invite you to submit a revised version of the manuscript that addresses the points raised during the review process.

There are two points that you need to pay special attention to:

Please take especially the issues raised by reviewer #2 seriously.

As reviewers #1 and #2 reported, there are non-substantiated deviations from the linked preregistration protocols to the manuscript, which is a serious issue (see also https://www.cos.io/initiatives/prereg, in addition, you might  refer  to PLoS ONEs rules for preregistered protocols, https://journals.plos.org/plosone/s/reviewer-guidelines#loc-reviewing-registered-reports). There are two ways how you can deal with this:  A.  You could remove the claims that your study is preregistered; alternatively:   B.  Carefully follow these rules:

1. Stick to the preregistered protocols where possible, and where not:

2. Substatiate in the manuscript why the Hypothesis or Method could not be followed; or alternatively, why the Hypothesis or Method was suboptimal and how the change represents an improvement.

3. Provide in your response letter a List of all cases where the protocols were not exactly followed, why, and how you made the change.

We look forward to receiving your revised manuscript.

Kind regards,

Marc H.E. de Lussanet, Ph.D.

Academic Editor

PLOS ONE

Journal Requirements:

2. We note that Figures 1 and 3 in your submission contain copyrighted images. All PLOS content is published under the Creative Commons Attribution License (CC BY 4.0), which means that the manuscript, images, and Supporting Information files will be freely available online, and any third party is permitted to access, download, copy, distribute, and use these materials in any way, even commercially, with proper attribution. For more information, see our copyright guidelines: http://journals.plos.org/plosone/s/licenses-and-copyright.

2.1.         You may seek permission from the original copyright holder of Figures 1 and 3 to publish the content specifically under the CC BY 4.0 license.

2.2.    If you are unable to obtain permission from the original copyright holder to publish these figures under the CC BY 4.0 license or if the copyright holder’s requirements are incompatible with the CC BY 4.0 license, please either i) remove the figure or ii) supply a replacement figure that complies with the CC BY 4.0 license. Please check copyright information on all replacement figures and update the figure caption with source information. If applicable, please specify in the figure caption text when a figure is similar but not identical to the original image and is therefore for illustrative purposes only.

Reviewers' comments:

Reviewer's Responses to Questions

**Comments to the Author**

1. Is the manuscript technically sound, and do the data support the conclusions?

Reviewer #1: Yes

Reviewer #2: Yes

Reviewer #3: Partly

2. Has the statistical analysis been performed appropriately and rigorously? 

Reviewer #1: Yes

Reviewer #2: Yes

Reviewer #3: I Don't Know

3. Have the authors made all data underlying the findings in their manuscript fully available?

Reviewer #1: Yes

Reviewer #2: Yes

Reviewer #3: Yes

4. Is the manuscript presented in an intelligible fashion and written in standard English?

Reviewer #1: Yes

Reviewer #2: Yes

Reviewer #3: Yes

5. Review Comments to the Author

Reviewer #1: Across two pre-registered experiments, this manuscript investigates threat perceptions based on body morphology. Findings were that bodies with larger mass were perceived as more threatening, even in the presence of facial information. Also, in Experiment 2, an interaction was found such that faces exerted more influenced with discordant bodies.

Overall, I quite enjoyed this paper. I found the methodology and analysis strong. The writing in the manuscript is clear and references the appropriate literature, and, for the most part, follows pre-registered methods (where there were some minor deviations, I believe these were improvements to what was pre-registered anyway).

The one minor suggest I have is that I found some of the Figures confusing, in particular, Figure 5 and Figure 6. For instance, could some colour be introduced in Figure 5 to help distinguish between the faces? Also, in Figure 6, it is unclear to me what is on the y-axis.

Reviewer #2: The manuscript describes two pre-registered experiments examining the relative influence of body and face morphology on perceptions of threat. Experiment 1 validates the approach with pre-selected faces designed to be high or low in threat and finds that participants do rate threatening faces as threatening. It also finds that body morphology also relates to threat, as more muscular or portly (ie. overall larger) bodies are rated as more threatening than emaciated (overall smaller) bodies. In Experiment 2, the authors show that these effects are not treated independently, as when faces and bodies are combined the respective threat levels are non-additively combined into a holistic perception of overall threat.

I thought this was a very good manuscript, clear and concise, that addressed an clearly articulated question with a thorough, competent methodology and appropriate analysis. The results are clear, which is impressive given how easily this paper could have become confusing. This is also my first time reviewing preregistered experiments, and I found it a positive experience (although I do have some methodological comments, see below). I see no reason that the study should not eventually be published, save for the following concerns that I hope the authors can address.

I found the hypotheses of Experiment 1 to be quite light – in particular, the hypotheses in the manuscript did not provide clear predictions about the different manipulations of body morphology that the authors actually used. Furthermore, these hypotheses appear to be different from the pre-registered hypotheses on the OSF registry, where the hypotheses and predictions are much clearer and more explicitly spelt out. I would encourage the authors to stick to the hypotheses as laid out in their preregistration – I also felt that these would follow the structure of the results section more intuitively.

The authors report an a priori G*Power analysis as justification of their sample size in Experiment 1 on p.5, but no such analysis is reported in the preregistration document, where the sample size is instead justified on the basis of previous research. An explanatory paragraph detailing inconsistencies with or deviations from the preregistration would be sufficient here On a related note, the authors have completely omitted to report the sample size of Experiment 2 (p.13).

Finally, it is not clear to me why the demographic data were collected. Participant height and BMI are explained (although glossed over in the results section as they consistently yield null results) but it is not clear why (in particular) age or education would be considered so important in the current study, and consequently it is difficult to know how to interpret the observed relationship between having a bachelors degree and finding muscular bodies threatening. Conversely, if the authors anticipated an interaction of attractiveness and participant gender, I am curious as to why information such as sexual orientation was not also gathered (I came back to this thought in the discussion on p.22 where the authors assert that male participants could not have found the bodies attractive in a romantic sense).

Some minor points about content

• P.11 – In the results of portliness, there are two instances where the authors instead refer to “the level of emaciation”

• P.13 – Regarding H6, it is not clear what the threat dimension of the face stimuli being ambiguous means. Specify that (presumably) this ambiguity refers to these faces being rated neutral.

• P.15 – I wonder if the authors would care to comment on the lower consensus on attractiveness ratings in Experiment 2 (ICC=.49) for the body-only stimuli relative to the same stimuli in Experiment 1 (ICC=.80). It seems to me that the compound stimuli may have served as influential companions – perhaps the presence of incompatible face/body compounds caused participants to change their expectations about the reliability of body cues.

• P.16 – No analysis of other variables (attractiveness, age, gender, etc.) is reported for the body-only stimuli of Experiment 2. If this is because this analysis yielded null results, these should still be reported

• Please label all axes of figures with relevant information – these should be interpretable to readers without having to mine the text (e.g. Fig 6, y axis)

• P.22 – The suggestion of using VR to display life-sized human stimuli is a nice idea, but you could also take this further by manipulating the size and body morphology of the participants’ own body avatar in VR to test additional predictions.

The following are minor stylistic points that reflect my personal preferences as a reader (and so can be used or disregarded as the authors see fit)

• P.4 – When reporting the results of previous studies in text using a numbered citation system, consider writing out the names of authors in the text; e.g. “For example, Schvey et al. [26] found that, in mock trials, male jurors…” instead of “For example, [26] found that, in mock trials, male jurors…”

• While they can be helpful when writing, abbreviations like PT and PA rarely actually help readers understand, particularly when these could be refer to simply as “threat” and “attractiveness”, which would be much easier to read

Reviewer #3: This paper reports how body morphological and facial elements affect the threat perception using CG images by adjusting the parameters of musculature, portliness, and emaciation in a stepwise manner.

The results show that the body morphological elements affect the threat perception as shown in previous studies and that there is also an interactive effect of the body morphology and facial traits. The finding is interesting in that it suggests that information about body size and the facial trait is comprehensively processed and influenced.

The work appears to be competently carried out. However, the paper is not convincing in that the format of tables/figures presentation, data analysis, and the lack of enough discussion to warrant publication. The paper needs to be revised.

1) First of all, the theme of this paper "threatening perception" seems to be based on the trait perception, but how does this differ from emotion perception? More profound reviews for previous studies and discussion for the current results would be better.

Also, in Fig 3, the size of the face itself seems to change depending on the parameters, which may need to be considered. The results reported indicate that age also affected the results. It is recommended to add a discussion of this point.

Furthermore, what is the role of the face in body morphology for threatening perception? It would be better to organize the discussion a little more to clarify your argument.

2) The lack of structured descriptions in the Methods and Results section gave the impression that the data were difficult and costly to interpret. Some of the analysis methods (e.g. intraclass correlation coefficients and logistic regression analysis) are common to both Experiment 1 and 2. Then, it would be better to describe them in a separate section "Data Analysis".

Also, for the logistic regression analysis, it would be desirable to have more detailed descriptions of the independent and dependent variables as well. They are likely to be necessary information to ensure reproducibility and to judge whether the statistical analysis is being appropriate.

As minor points:

1) There were several areas where the literature was not properly cited. (e.g. the Introduction section [26]-[28] in p.4) Please recheck the citation format.

2) Recheck the captions in the tables S5-S8 to make sure they are correct. (AIC value is smaller in other Models.)

3) Tables and figures are requested to be placed in appropriate and effective places in your paper. It would be unfavourable to list them all in one place.

4) The way "H1" might mean "Hypothesis 1"(e.g. p.5), however, it may not be a universal usage. Other abbreviations such as IV was first mentioned (p.9), so it would be better to pay more attention to the use of abbreviations.

5) How many participants joined in Experiment 2 finally?

6) It would be better to elaborate more on the relationship between the number of stimuli in Experiment 2 to the number of trials? (Does it include the number of distractions?)

7) It would be better to describe the dataset of the face stimuli in more detail. What other parameters are in the dataset besides "threatening"? It would be better to write in a way that the reader can understand a summary of the presented stimuli without referencing the research by Todorov et al. (2013) [33].

6. PLOS authors have the option to publish the peer review history of their article (what does this mean?). If published, this will include your full peer review and any attached files.

Reviewer #1: No

Reviewer #2: No

Reviewer #3: No

---

## [Author Response · Author response to Decision Letter 0]

15 Jan 2021

Response to the editor

We would like to thank the Editor and reviewers for their constructive suggestions on our manuscript entitled “Perceiving threat in others: the role of body morphology”. Based on their comments we have revised our manuscript accordingly. The most substantial revisions relate to deviations from the pre-registered protocol in Experiment 1, as highlighted by both the Editor and Reviewer 2.

The manuscript deviated from the pre-registration in three main ways. First, the pre-registration contained a greater number of individual hypotheses. The submitted manuscript had collapsed these into broader, more concise hypotheses. These have since been restored in the revised manuscript to the original number of hypotheses to match the pre-registration. Second, the manuscript described an a priori G*Power analysis to justify the sample size of Experiment 1. This was done following the publishing of the pre-registration, and so should not have been included. This has now been replaced by a post-hoc estimation of achieved power. Third, the pre-registered protocol had outlined the use of linear models to analyse perceived threat in the face and body stimuli. We later decided that, as ordinal scales had been used to record perceived threat, it would be more apt to employ a model used specifically for this type of data. This has now been clarified in the revised manuscript, where a detailed justification for this change is included in the new “Data Analysis” subsection found at the conclusion to the Methods section. 

In addition, as per request, Figure 3, which depicted examples of the compound stimuli, has been removed such as to comply with the Creative Commons Attribution License (CCAL), CC BY 4.0. Therefore, the numbering for each following figure has been updated. The paper also now more closely follows PLOS ONE’s style requirements. Below we address these, and all other concerns raised by the three reviewers.

Response to reviewers

We thank the reviewers for reviewing our manuscript “Perceiving threat in others: the role of body morphology” and for their constructive comments. Below we include a point by point reply to the issues raised by each reviewer. 

Reviewer 1

The one minor suggestion I have is that I found some of the Figures confusing, in particular, Figure 5 and Figure 6. For instance, could some colour be introduced in Figure 5 to help distinguish between the faces? Also, in Figure 6, it is unclear to me what is on the y-axis.

- We agree that Figures 5 and 6 (now Figures 4 and 5 in the updated manuscript due to the removal of Figure 3) could have been made clearer. These have been revised accordingly and Figure 5 now includes different colours to distinguish the varying levels of facial threat and musculature of the stimuli. We have improved the y-axis label of Figure 6 to increase clarity.

Reviewer 2

I found the hypotheses of Experiment 1 to be quite light – in particular, the hypotheses in the manuscript did not provide clear predictions about the different manipulations of body morphology that the authors actually used. Furthermore, these hypotheses appear to be different from the pre-registered hypotheses on the OSF registry, where the hypotheses and predictions are much clearer and more explicitly spelt out. I would encourage the authors to stick to the hypotheses as laid out in their preregistration – I also felt that these would follow the structure of the results section more intuitively.

- We originally condensed the hypotheses used in the pre-registration to improve the readability of the manuscript. However, this did leave the individual hypotheses lighter in detail and not as clearly structured as they had originally appeared in the pre-registration. In the revised manuscript, each body manipulation now has its own separate numbered hypothesis with justification, matching the pre-registration (see pages 5/6 of the revised manuscript). These numbered hypotheses are now also directly referred to in appropriate subsections of the Results. 

Also, the first pre-registered hypothesis of Experiment 2 (concerning the ICCs for perceived threat) had not been explicitly stated in the text. This is now clearly stated (see page 14).

The authors report an a priori G*Power analysis as justification of their sample size in Experiment 1 on p.5, but no such analysis is reported in the preregistration document, where the sample size is instead justified on the basis of previous research. 

- This was an error on our part. This a priori G*Power analysis was done following the publication of the pre-registration, and so should not have been included in this pre-registered paper. This has now been replaced by a post-hoc estimation of achieved power (see page 6).

The authors have completely omitted to report the sample size of Experiment 2 (p.13).

- We thank the reviewer for pointing this out. This has been amended in the revised manuscript (page 15).

It is not clear to me why the demographic data were collected. Participant height and BMI are explained (although glossed over in the results section as they consistently yield null results) but it is not clear why (in particular) age or education would be considered so important in the current study, and consequently it is difficult to know how to interpret the observed relationship between having a bachelors degree and finding muscular bodies threatening.

- We thank the reviewer for pointing this out. We agree that justification of the inclusion of the age and education variables was limited and lacked further discussion. As these were not key variables of interest, but rather control variables, the original manuscript did not go into much detail. The revised manuscript addresses this at the conclusion of the Introduction by more clearly illustrating why they were included as controls (page 6). Furthermore, the Discussion section now includes a paragraph that briefly discusses the observed relationships between age, education and perceived threat (page 26).

Conversely, if the authors anticipated an interaction of attractiveness and participant gender, I am curious as to why information such as sexual orientation was not also gathered.

- We thank the reviewer for this comment. This is a valid limitation, which is now noted in the Discussion of the revised manuscript (page 25).

Minor content comments:

1) P.11 – In the results of portliness, there are two instances where the authors instead refer to “the level of emaciation” 

- Thank you, this has been fixed.

2) Regarding H6, it is not clear what the threat dimension of the face stimuli being ambiguous means. Specify that (presumably) this ambiguity refers to these faces being rated neutral.

- Thank you, more detail has now been added in the text to make this clearer (page 14).

3) P.15 – I wonder if the authors would care to comment on the lower consensus on attractiveness ratings in Experiment 2 (ICC=.49) for the body-only stimuli relative to the same stimuli in Experiment 1 (ICC=.80).

- Thank you, we agree that this fall in consensus is most likely attributable to their presentation in the same block as the compound stimuli. This is now addressed in the text (page 18).

4) P.16 – No analysis of other variables (attractiveness, age, gender, etc.) is reported for the body-only stimuli of Experiment 2. If this is because this analysis yielded null results, these should still be reported.

- In the original manuscript, it was unclear as to how the models reported in the text were selected. We have now clarified that these were selected using the Akaike Information Criterion (AIC). For the body-only stimuli in Experiment 2, the best-fitting model was the initial model, in which threat was regressed onto only the changes in musculature. The other variables had no significant effect, and disimproved the fit of the model, and so were not reported. This has now been made clear in the text (page 19)

5) Please label all axes of figures with relevant information – these should be interpretable to readers without having to mine the text (e.g. Fig 6, y axis)

- Thank you, this has been changed.

6) P.22 – The suggestion of using VR to display life-sized human stimuli is a nice idea, but you could also take this further by manipulating the size and body morphology of the participants’ own body avatar in VR to test additional predictions.

- This is a really nice suggestion, thank you. We have added it to the discussion (page 26).

Minor stylistic comments:

P.4 – When reporting the results of previous studies in text using a numbered citation system, consider writing out the names of authors in the text; e.g. “For example, Schvey et al. [26] found that, in mock trials, male jurors…” instead of “For example, [26] found that, in mock trials, male jurors…”

- Thank you, this has been changed.

While they can be helpful when writing, abbreviations like PT and PA rarely actually help readers understand, particularly when these could be refer to simply as “threat” and “attractiveness”, which would be much easier to read

- These abbreviations have now been largely removed, aside for some uses in figure axes.

Reviewer 3

The theme of this paper "threatening perception" seems to be based on the trait perception, but how does this differ from emotion perception? More profound reviews for previous studies and discussion for the current results would be better.

- Thank you, the difference between emotion and trait perception is a key distinction for the current paper. This distinction has now been highlighted and clarified in the Introduction (see page 3 of the revised manuscript). Also, a more expanded paragraph on the link between emotion and trait perception has been added to the discussion (see page 24).

In Fig 3, the size of the face itself seems to change depending on the parameters, which may need to be considered

- The sizes of the faces in the compounds were slightly adjusted to match the size of the paired bodies more naturally. Although this may have introduced some slight variability in how the faces were perceived, it was decided that this was preferable to presenting noticeably incongruent face and body combinations. This adjustment of face size has now been clarified in the compound stimuli subsection of the Methods section of Experiment 2 (page 16).

The results reported indicate that age also affected the results. It is recommended to add a discussion of this point.

- We thank the reviewer for their comment. As also noted by Reviewer 2, the analyses of the effects of age and education was lacking further discussion. The revised manuscript addresses this, with the Discussion section now including a paragraph that briefly discusses the observed relationships between age, education and perceived threat (page 26).

What is the role of the face in body morphology for threatening perception? It would be better to organize the discussion a little more to clarify your argument.

- We understand that we may not have fully clarified the primary role played by facial information in the perception of threat. Although we have shown that bodies play a significant role, regardless of facial information, it does seem that the judgment is still primarily driven by the face. This has now been made clear in the opening of the Discussion (page 23).

The lack of structured descriptions in the Methods and Results section gave the impression that the data were difficult and costly to interpret. Some of the analysis methods (e.g. intraclass correlation coefficients and logistic regression analysis) are common to both Experiment 1 and 2. Then, it would be better to describe them in a separate section "Data Analysis". Also, for the logistic regression analysis, it would be desirable to have more detailed descriptions of the independent and dependent variables as well. They are likely to be necessary information to ensure reproducibility and to judge whether the statistical analysis is being appropriate.

- We appreciate that the results may be difficult to follow because we have a number of analyses to present. However, both Reviewers 1 & 2 indicated that they thought the results and analyses were particularly clear, so we have tried to clarify sections without substantially changing the structure. To achieve this, we followed the reviewer’s suggestion and have added a “Data Analysis” subsection to the end of each Methods section (pages 9/17). This outlines the analysis steps due to follow in each subsequent Results section. We also now include details of the ICC and regression analyses that were previously found in the Results section. 

Minor points:

1) There were several areas where the literature was not properly cited. (e.g. the Introduction section [26]-[28] in p.4) Please recheck the citation format.

- Thank you, this has been fixed.

2) Recheck the captions in the tables S5-S8 to make sure they are correct. (AIC value is smaller in other Models.)

- The reported models are from the best-fitting models that are also fully-powered. These models do not include height/BMI, as 31 participants opted not to disclose this information.

3) Tables and figures are requested to be placed in appropriate and effective places in your paper. It would be unfavourable to list them all in one place

- We thank the reviewer for the suggestion. Figures now appear in the manuscript at the conclusion of the paragraph in which they were first mentioned. Although regression tables could have been included in the manuscript itself, we decided to omit them from the main text. We felt that plotting out the relationships visually would communicate the relationships between the key variables more clearly. 

4) The way "H1" might mean "Hypothesis 1"(e.g. p.5), however, it may not be a universal usage. Other abbreviations such as IV was first mentioned (p.9), so it would be better to pay more attention to the use of abbreviations.

- Thank you, all abbreviations have now been clarified at their first use in the text.

5) How many participants joined in Experiment 2 finally?

- We thank the reviewer for pointing this omission. This has been amended in the revised manuscript (page 15).

6) It would be better to elaborate more on the relationship between the number of stimuli in Experiment 2 to the number of trials? (Does it include the number of distractions?)

- The exact number of trials for experimental, distractor, body and compound stimuli in Experiment 2 is now explicitly stated in the text (page 17).

7) It would be better to describe the dataset of the face stimuli in more detail. What other parameters are in the dataset besides "threatening"? It would be better to write in a way that the reader can understand a summary of the presented stimuli without referencing the research by Todorov et al. (2013) [33].

- More detail has now been added to the Methods section to more fully describe the method used by Todorov to produce the CG face stimuli (page 7).

---

## [Decision Letter · Decision Letter 1]

16 Feb 2021

PONE-D-20-19322R1

Perceiving threat in others: the role of body morphology

PLOS ONE

Dear Dr. McElvaney,

Thank you for submitting your manuscript to PLOS ONE. After careful consideration, we feel that it has merit but does not fully meet PLOS ONE’s publication criteria as it currently stands. Therefore, we invite you to submit a revised version of the manuscript that addresses the points raised during the review process.

We look forward to receiving your revised manuscript.

Kind regards,

Marc H.E. de Lussanet, Ph.D.

Academic Editor

PLOS ONE

Reviewers' comments:

Reviewer's Responses to Questions

**Comments to the Author**

1. If the authors have adequately addressed your comments raised in a previous round of review and you feel that this manuscript is now acceptable for publication, you may indicate that here to bypass the “Comments to the Author” section, enter your conflict of interest statement in the “Confidential to Editor” section, and submit your "Accept" recommendation.

Reviewer #2: (No Response)

Reviewer #3: (No Response)

2. Is the manuscript technically sound, and do the data support the conclusions?

Reviewer #2: Yes

Reviewer #3: Yes

3. Has the statistical analysis been performed appropriately and rigorously? 

Reviewer #2: Yes

Reviewer #3: Yes

4. Have the authors made all data underlying the findings in their manuscript fully available?

Reviewer #2: Yes

Reviewer #3: Yes

5. Is the manuscript presented in an intelligible fashion and written in standard English?

Reviewer #2: Yes

Reviewer #3: Yes

6. Review Comments to the Author

Reviewer #2: The authors have addressed my major concerns to my satisfaction, and I have no further reservations about recommending the manuscript for publication. I also appreciate the changes made in response to the other reviewers, particularly the effort made to relate the current study to the emotion perception literature.

Regarding the power analysis, I agree that the use of an a priori analysis is not appropriate if it was performed following data collection. If it was done between preregistration and data collection, this would be fine but would require some explanation—the purpose of preregistration should not be to constrain experimenters in running a study effectively or prohibit them from going ‘off script’, but to maximise transparency about the motivations and timing of decisions that can affect one’s confidence in the results and conclusions.

For future reference, I understand that post-hoc calculations of observed power are considered somewhat problematic (see http://daniellakens.blogspot.com/2014/12/observed-power-and-what-to-do-if-your.html). Given that the authors do report an a priori justification for their sample size in both their preregistration and manuscript, I would drop this post-hoc calculation. If a power calculation is required, I would recommend looking into simulation-based approaches.

Minor point:

p.9 – when first defining abbreviations, spell out the full abbreviation (i.e. ‘intraclass correlation coefficient (ICC)’ rather than just correlation coefficient).

As far as I can tell from the supplementary material, the AIC was used to select the best fitting model in both experiments, but is only mentioned in Experiment 2 because it selected the simpler model over the full. If this model comparison was conducted, it should be reported in the main text for Experiment 1 as well.

Reviewer #3: I am pleased with the way the authors have addressed the issues I raised. The revised manuscript has a more organized structure, making it much easier to understand. I would like to add a few more comments.

Introduction

- Thank you for adding the description of emotion perception and trait perception. It helped me understand the points. However, there is another ambiguous term: personality. I am curious about the difference between personality and trait in the manuscript. It would be helpful to add some descriptions of personality and traits.

Furthermore, which process is more similar to "perceiving threat": "trait perception" or "emotion perception"? I've regarded "perceiving threat", examined in this study, as a part of judging the traits of a person presented as a stimulus. However, now I feel that it contains both aspects: trait perception and emotion perception. In the last paragraph on p.4, the authors argued that perceiving threat might influence the judge of traits or personality. Therefore, it would be more persuasive if the authors clarify the position on whether they regard “perceiving threat” as more like “trait perception” or “emotion perception”.

- In the last hypothesis on p5, the authors refer to the holistic processing of faces and bodies and hypothesized that faces and bodies would play significant roles respectively. This hypothesis sounds very general, and it would be good to add some more description of the holistic person-perception hypothesis given in previous works. This is also mentioned in the Discussion section (p.26) and would be a key point in the manuscript. For a better understanding of the authors’ hypothesis about perceiving threat, it would be useful to add more explanations of the reference [10].

Methods

- Am I correct in understanding that the final number of participants was 150 for experiment 1 (p.7), of which 87 were women, and 100 for experiment 2, of which 74 were women (p.16-17)? It would be clear for readers if the authors clarify the total number of participants.

Results

Exp1

- It is a trivial matter, but in hypothesis H5 and H6, the order was Portliness (H5) and Emaciation (H6) (p.5). Therefore, it would be helpful if the order of the results (p.13-15) match with the order written in the hypothesis, for a smoother understanding.

Exp2

- Regarding the last paragraph in section 3.2.2.3 on p. 22, the authors argue that contrary to H7 and H8, there was no correlation between the musculature or facial dimension and the compound stimuli ratings. However, I'm not sure why this conclusion was drawn. For example, for H7, I thought it needed to compare the correlation coefficients when the face was ambiguous (i.e., when the face dimension level was about 1-3) and when the face dimension level was other levels. However, only overall correlation coefficients have been reported, and is it possible to draw such a conclusion? I apologize if I misunderstand the results.

Discussion

- As I mentioned above, holistic processing is the key to the authors’ hypothesis and interpretation of the results, but the explanation may be insufficient. So, I cannot fully understand how the results supported the holistic person-perception hypothesis (p.26). It would be helpful if the authors add more explanations of the holistic person-perception hypothesis in the Introduction section and rephrased it shortly in the Discussion section.

7. PLOS authors have the option to publish the peer review history of their article (what does this mean?). If published, this will include your full peer review and any attached files.

Reviewer #2: No

Reviewer #3: No

---

## [Author Response · Author response to Decision Letter 1]

26 Feb 2021

Response to the editor

We would like to thank the Editor and reviewers for their constructive suggestions on our manuscript entitled “Perceiving threat in others: the role of body morphology”. Based on their comments we have revised our manuscript accordingly. The main revisions relate to queries by Reviewer 3 regarding expansions on the holistic processing hypothesis and additional detail on the distinction between emotion and trait perception. Below we address these, and all remaining concerns raised by the reviewers.

Response to reviewers

We thank the reviewers for reviewing our manuscript “Perceiving threat in others: the role of body morphology” and for their constructive comments. Below we include a point by point reply to the issues raised by each reviewer. 

Reviewer 2

Regarding the power analysis, I agree that the use of an a priori analysis is not appropriate if it was performed following data collection. Given that the authors do report an a priori justification for their sample size in both their preregistration and manuscript, I would drop this post-hoc calculation. If a power calculation is required, I would recommend looking into simulation-based approaches.

- We thank the reviewer for this comment. We agree that post-hoc power calculations can often be more problematic than helpful. Also, as the reviewer has mentioned, we did originally report an a priori justification for our sample size in both the preregistration and manuscript.

Therefore, as suggested, we have removed the post-hoc power calculations from the manuscript. 

p.9 – when first defining abbreviations, spell out the full abbreviation (i.e. ‘intraclass correlation coefficient (ICC)’ rather than just correlation coefficient).

- Thank you, this has now been fixed (page 9). 

As far as I can tell from the supplementary material, the AIC was used to select the best fitting model in both experiments, but is only mentioned in Experiment 2 because it selected the simpler model over the full. If this model comparison was conducted, it should be reported in the main text for Experiment 1 as well.

- Thank you, the use of the AIC is now reported in the Data Analysis sections of the Methodology for both Experiments 1 (page 9) and Experiment 2 (page (17) 

Reviewer 3

Thank you for adding the description of emotion perception and trait perception. It helped me understand the points. However, there is another ambiguous term: personality. I am curious about the difference between personality and trait in the manuscript. It would be helpful to add some descriptions of personality and traits. 

- Thank you, we now appreciate that we have used the terms “personality” and “trait” somewhat interchangeably. When we have referred to “trait perception”, the full term we are referring to is “personality trait perception”. We have now clarified this with use of the full term in the early parts of the introduction section (pages 3/4). 

Furthermore, which process is more similar to "perceiving threat": "trait perception" or "emotion perception"? I've regarded "perceiving threat", examined in this study, as a part of judging the traits of a person presented as a stimulus. However, now I feel that it contains both aspects: trait perception and emotion perception. In the last paragraph on p.4, the authors argued that perceiving threat might influence the judge of traits or personality. Therefore, it would be more persuasive if the authors clarify the position on whether they regard “perceiving threat” as more like “trait perception” or “emotion perception”.

- We thank the reviewer for this comment. We agree that this is a very important distinction to draw. An emotional signal can indeed influence how a person is perceived. However, emotions tend to be transient in nature. Hence, the goal of the current paper is to go beyond inferences that may be reliant on, or driven by, perceived emotion. Therefore, for the purposes of the current paper, we are focusing on threat as a perceived trait, rather than as a perceived emotion. We have further specified the distinction between emotion and trait perception, and more clearly stated our goal of investigating personality trait inferences, in the manuscript (page 3).

- In the last hypothesis on p5, the authors refer to the holistic processing of faces and bodies and hypothesized that faces and bodies would play significant roles respectively. This hypothesis sounds very general, and it would be good to add some more description of the holistic person-perception hypothesis given in previous works. This is also mentioned in the Discussion section (p.26) and would be a key point in the manuscript. For a better understanding of the authors’ hypothesis about perceiving threat, it would be useful to add more explanations of the reference [10].

- We thank the reviewer for their comment. We have provided additional information about the nature of the holistic processing hypothesis forwarded by Aviezer et al. [10], along with more detail about the experiments put forth in their paper as evidence for the hypothesis (page 3). 

Am I correct in understanding that the final number of participants was 150 for experiment 1 (p.7), of which 87 were women, and 100 for experiment 2, of which 74 were women (p.16-17)? It would be clear for readers if the authors clarify the total number of participants.

- Thank you, this has now been clarified in the manuscript.

It is a trivial matter, but in hypothesis H5 and H6, the order was Portliness (H5) and Emaciation (H6) (p.5). Therefore, it would be helpful if the order of the results (p.13-15) match with the order written in the hypothesis, for a smoother understanding.

- Thank you for this helpful suggestion, the hypotheses have now been reordered to match the layout of the results section. 

Regarding the last paragraph in section 3.2.2.3 on p. 22, the authors argue that contrary to H7 and H8, there was no correlation between the musculature or facial dimension and the compound stimuli ratings. However, I'm not sure why this conclusion was drawn. For example, for H7, I thought it needed to compare the correlation coefficients when the face was ambiguous (i.e., when the face dimension level was about 1-3) and when the face dimension level was other levels. However, only overall correlation coefficients have been reported, and is it possible to draw such a conclusion? I apologize if I misunderstand the results.

- Thank you, we see now that this was not very clearly worded. We did not mean to suggest that no correlations were found between perceived threat in the face/body stimuli and compound stimuli. This was not the case, as threat in the compounds was significantly correlated with all levels of face-only and body-only threat. Rather, no clear pattern in correlation by level of musculature or facial dimension emerged, which was contrary to H7 and H8. We have now clarified this distinction in the manuscript (page 21).

As I mentioned above, holistic processing is the key to the authors’ hypothesis and interpretation of the results, but the explanation may be insufficient. So, I cannot fully understand how the results supported the holistic person-perception hypothesis (p.26). It would be helpful if the authors add more explanations of the holistic person-perception hypothesis in the Introduction section and rephrased it shortly in the Discussion section

- As suggested, more information about the hypothesis has been provided in the Introduction, with an additional sentence in the Discussion (page 24) detailing how the perception of the compounds diverged from a simple linear combination of the threat perceived in the face and body stimuli.

---

## [Editor Report · Decision Letter 2]

25 Mar 2021

Perceiving threat in others: the role of body morphology

PONE-D-20-19322R2

Dear Dr. McElvaney,

We’re pleased to inform you that your manuscript has been judged scientifically suitable for publication and will be formally accepted for publication once it meets all outstanding technical requirements.

Kind regards,

Marc H.E. de Lussanet, Ph.D.

Academic Editor

PLOS ONE
---

## [Editor Report · Acceptance letter]

29 Mar 2021

PONE-D-20-19322R2 

Perceiving threat in others: the role of body morphology 

Dear Dr. McElvaney:

I'm pleased to inform you that your manuscript has been deemed suitable for publication in PLOS ONE. Congratulations! Your manuscript is now with our production department. 

Kind regards, 

on behalf of

Dr. Marc H.E. de Lussanet 

Academic Editor

PLOS ONE